# Tuning the Hydration Acceleration Efficiency of Calcium Carbonate by Pre-Seeding with Calcium Silicate Hydrate

**DOI:** 10.3390/ma15196726

**Published:** 2022-09-28

**Authors:** Han Yan, Dongliang Zhou, Yong Yang, Xin Shu, Cheng Yu, Qianping Ran

**Affiliations:** 1State Key Laboratory of High Performance Civil Engineering Materials, Nanjing 211103, China; 2School of Material Science and Engineering, Southeast University, Nanjing 211189, China

**Keywords:** cement, hydration, calcium silicate hydrate, composite, calcium carbonate

## Abstract

Nanomaterials are promising candidates for refined performance optimization of cementitious materials. In recent years, numerous studies about the performance improvement of nanomaterials using polymers have been conducted, but the modification of cement-oriented nanomaterials with inorganic modifiers is seldom assessed. In this study, we explored the performance tuning and optimization of nanomaterials by inorganic modification. In this work, hydration acceleration efficiency of calcium carbonate (CaCO_3_, CC) was tuned via surface deposition with calcium silicate hydrate (C–S–H) nanogel through seeding. Multiple calcium carbonate–calcium silicate hydrate (CC–CSH) samples with varying degrees of surface modification were prepared via dosage control. According to characterizations, the degree of C–S–H modification on the CaCO_3_ surface has a maximum that is controlled by available surface space. Once the available space is depleted, excessive C–S–H turns into free form and causes adhesion between CC–CSH particles. The resultant CC–CSH samples in this work showed enhanced hydration acceleration efficiency that is tuned by the actual degree of C–S–H modification. Elevated C–S–H modification causes CC–CSH’s acceleration behavior to shift to enhanced early-age acceleration. According to mortar strength tests, CC–CSH with 5% C–S–H modification showed the most balanced performance, while CC–CSH with higher C–S–H modification showed faster early-age strength development at the cost of lower later-age strength. The inferior later-age strength of highly C–S–H-modified CC–CSH samples may be due to the coarsening of hydration products and stiffening of their network, as well as agglomeration caused by C–S–H adhesion. This study may offer a novel route for performance tuning of cement-oriented nanomaterials.

## 1. Introduction

Cementitious material is one of key materials in construction and buildings [1]. In past decades, research on cementitious material has focused on refined performance optimization [2] and sustainability of development [3]. Recently, nanomaterials are enjoying widening applications as admixtures or additives in optimizing the performance of cementitious materials [2,4,5,6,7]. In cement and concrete, various nanomaterials show application potential in hydration acceleration [6,8,9,10,11,12,13], strength enhancement [14,15], permeability reduction [16,17], durability improvement [18,19,20], etc., with noteworthy performance and few adverse consequences. The application of nanotechnology for cementitious materials may help relieve limitations of current performance enhancing techniques, such as strength loss and durability deterioration at later ages [19,21]. Furthermore, in other concerning aspects, such as sulfate balance [22], soundness [23], and alkali-activated systems [24], the application potential of nanomaterials is also promising.

Hydration acceleration is one of the major application aspects of nanomaterials, as hydration is the key chemical reaction of Portland cement [1]. The acceleration effect is usually achieved by seeding. In past decades, extensive studies concerning numerous nanomaterials, such as nanoSiO_2_ [7,8,15], synthetic nano calcium silicate hydrate (C–S–H) [9,12], and nano CaCO_3_ [10,13], have been carried out. These materials exhibited prominent hydration acceleration efficiency compared to traditional materials, with few adverse effects on later-age index [4].

Still, there are several problems in the application of nanomaterials in this regard. At present, the effect of nanomaterials is mainly adjusted by dosage and mix-compounding [5], just like conventional admixtures. However, due to their high interfacial activity, and hence, proneness to agglomerate in pore solution [25], boosting the dosage may bring about undesired agglomeration, causing efficiency loss and workability deterioration. These adverse effects are further observed as dosage sensitivity and poor performance reproducibility [2,7]. Instead of dosage adjustment, modifying or tuning the intrinsic properties of nanomaterials may offer alternative solutions. In recent years, there have been extensive studies on optimizing the effect of nanomaterials in cementitious materials with surface modification [26,27]. Still, these modifications are mainly achieved by polymers, and the use of inorganic modifiers is seldom reported. 

In this work, performance tuning of calcium carbonate (CaCO_3_, abbrev. CC), a common nanomaterial frequently used as a hydration accelerator, with inorganic modification, was investigated. The inorganic modification was conducted via surface deposition on calcium carbonate with calcium silicate hydrate (C–S–H). This idea comes with the acceleration mechanism of CaCO_3_, which is achieved by seeding C–S–H [28]. By pre-conducting the C–S–H seeding process on CaCO3 nanoparticles, i.e., activating CaCO3 with nano C–S–H, a tunable hydration acceleration effect can be achieved. We prepared a series of CaCO3 nanoparticles with different degrees of C–S–H activation in this study. The modified CaCO_3_ nanoparticles, namely, CaCO3-C–S–H (CC–CSH), underwent cement-related tests and characterizations to investigate the effect of C–S–H modification and the viability of nanomaterial modification in cementitious systems.

## 2. Materials and Methods

### 2.1. Materials

CaCO_3_ nanoparticles were supplied by Changzhou CaCO_3_ Co. Ltd. (Changzhou, China). The particles were produced by carbonation, with no surface modification. The mean diameter of the particles was 1.6 μm (by surface area). Other reagents for CC–CSH preparation, namely, calcium nitrate tetrahydrate (Ca(NO_3_)_2_ 4H_2_O, A.R.) and sodium silicate (Na_2_SiO_3_ 9H_2_O, A.R.), were purchased from Sinopharm Co. Ltd., Shanghai, China. A polycarboxylate ether dispersant (PCE) was used to improve the stability of CC–CSH dispersion, supplied by Nanjing Bote Co. Ltd. (Nanjing, China) in the form of 40% solution. PCE was used as the superplasticizer in cement-related experiments. The chemical structure of PCE is shown in Figure 1.

P I 42.5 (based on GB/T 175-2007 [29]) cement was used in this study; its mineral content is listed in Table 1.

### 2.2. Preparation and Characterization of CC–CSH

#### 2.2.1. Preparation of CC–CSH

CC–CSH was prepared by depositing nano C–S–H onto the surface of CaCO_3_. The mechanism simulates the seeding process that occurs in CaCO_3_-modified cements and is based on findings by Bentz et al. [28]. CC–CSH with different C–S–H content was prepared. In the preparation process, 100.0 g of CaCO_3_ and PCE solution containing 5.00 g of the dispersant were dosed in 1000 mL of distilled water and stirred. Then, 200 mL of Ca(NO_3_)_2_ 4H_2_O solution and equivalent volume of Na_2_SiO_3_ 9H_2_O solution were added to the CaCO_3_ suspension under continuous stirring via a peristaltic pump at a constant rate. The dosage of Ca(NO_3_)_2_ 4H_2_O and Na_2_SiO_3_ 9H_2_O is listed in Table 2 and was determined based on C–S–H content. A CaCO_3_ reference without C–S–H deposition (but with PCE addition) was also prepared.

The reaction was carried out at room temperature, and the addition of both solutions was completed simultaneously in 12 h. The final dispersion was dialyzed until becoming NO_3_^−^ negative (<10 ppm, measured by ionic chromatography, TOSOH IC-2010, TOSOH Co., Tokyo, Japan) and stored for further use.

#### 2.2.2. Characterization of CC–CSH

The mean size of CC–CSH was determined by a HELOS-SUCELL Size Analyzer (Sympatec Co., Clausthal-Zellerfeld, Germany). Free nano C–S–H in CC–CSH dispersion was sampled by short-term (5 min) centrifugation at 3000 rpm, in which CC–CSH was separated while free nano C–S–H remained in the supernatant. This method was also used for free C–S–H removal in subsequent experiments. Its size was measured by dynamic light scattering (DLS, type CGS-3 ALV Co., Langen, Germany). 

Shape and morphology, as well as the presence of C–S–H on CaCO_3_, were then observed by a Scanning Electron Microscope–Energy-Dispersive Spectrometer (SEM-EDS, type QUANTA 250, FEI Co., Hillsboro, OR, USA); the acceleration voltage was 15 kV. Free C–S–H was removed using the technique mentioned above and CC–CSH was diluted with a proper amount of distilled water, dropped onto sample platelets, and dried at room temperature for 24 h before observation.

### 2.3. Preparation of Mortar and Paste Samples

#### 2.3.1. Mortar Preparation

Cement mortar mixes were prepared according to GB/T 17671-1999 [30] with a water to binder (including cement and CaCO_3_/CC–CSH) ratio (w/b) of 0.4 and a binder (total 600 g) to sand (1350 g) ratio of 4/9 by weight. The mix design of the cement composites is demonstrated in Table 3. PCE of 0.12–16% total binder mass was used to regulate the flow of mortar to 180 ± 10 mm.

Mortars were mixed on a Jianyi JJ-5 mortar mixer. Each mortar sample was prepared for three repetitions of each test. The strength was tested at 12 h, 1 d, 7 d, and 28 d. The prisms were demolded after 10–11 h and cured in a Dongwu HBY-40B curing case at 20 ± 0.5 °C until testing time.

#### 2.3.2. Paste Preparation

The pastes with w/b of 0.4 were prepared in a high-shear mixer (Jianyi NJ-160A, Jianyi Co., Nanjing, China) at room temperature (20 °C). Superplasticizer (SP, polycarboxylates) and CC–CSH were added to the water phase before mixing with cement (296.4 g). The amount of CC–CSH added was 3.6 g (in solid, counted as binder), 1.2% by total binder mass, and the amount of PCE was 0.075–0.10% binder mass (regulating flow to 180 ± 10 mm). In the mixing process, the total 300 g of binder was mixed with the water phase at a low speed for 2 min. Then, the mix was paused for 15 s and then resumed at a high speed for another 2 min. 

The paste samples that were not involved in fresh-state characterizations were cured for the desired time at 20 °C in plastic vials, then demolded. The outer surface (1 mm thickness) of samples was removed using a Buehler Phoenix 4000 plate smoothing grinder to discard carbonated parts.

### 2.4. Testing and Characterization of the Mortar and Paste Samples

#### 2.4.1. Mortar Strength Tests

Flexural and compressive strength of the mortar samples were also tested according to GB/T 17671-1999 [30]. Both compressive and flexural strength were assessed using an Aelikon AEC-201 test machine.

The strength development of cement mortars with the CC–CSH samples was tested. After this test, the dosage dependence of CC–CSH was measured using CC–CSH, which has the most balanced performance. Dosages of 0.3%, 0.6%, 1.2%, and 2.0% (vs. total binder mass) were assessed.

#### 2.4.2. Microscopic Observations

Hardened paste samples were crushed, and slices of 3~5 × 0.5~1.5 mm were gathered for observation. SEM observation was conducted on the FEI Quanta 250 Scanning electron microscope (FEI Co., Hillsboro, OR, USA) at a magnification of 10,000×.

#### 2.4.3. Isothermal Calorimetry (IC) Tests

In IC tests, about 14.00 g of the paste (prepared via 2.4.1) was quickly and accurately weighed into a plastic vial. The vial was sealed and placed in a TAM Air isothermal calorimeter to measure heat development in 36 h at 20.0 °C. Since the mixing and weighing processes were conducted outside, the early hydration of all samples was not measured for about 5 min from the starting time of mixing. Due to the limited number of testing slots, the pastes were prepared here with 1.2% CaCO_3_ and CC–CSH (in solid) replacement.

## 3. Results

### 3.1. Characteristics of CC–CSH Particles

The mean size and free C–S–H content of CC–CSH samples are shown in Table 4. As the results suggest, for CC–CSH with a relatively low degree of C–S–H modification, most C–S–H was fixed on CaCO_3_ particles, and only few existed as nanogels. For CC–CSH-20, most C–S–H was in free form, which is due to the depletion of available CaCO_3_ surfaces. Taking out free C–S–H, the actual modification degrees of CC–CSH-10 and CC–CSH-20 were 7.73% and 9.38%, respectively. As shown in Table 4, the size of free nano C–S–H was much smaller than the CaCO_3_ particles, so the size of CC–CSH with 5% C–S–H modification was nearly the same as unmodified CaCO_3_. However, CC–CSH-10 and CC–CSH-20 were notably larger, which can be attributed to the adhesion effect of C–S–H [31] (as Figure 1 shows). The effect became more prominent with higher C–S–H contents, resulting in the agglomeration of a larger fraction of CC–CSH.

The SEM images of the CC–CSH samples are shown in Figure 2. As observed, the surfaces of unmodified CaCO_3_ were rather smooth, with varied size ranging from less than 1 μm to 5 μm. After modification with C–S–H, the surface of the calcium carbonate particles became rougher. The surface morphology of the samples showed resemblance to the C–S–H grown in slightly hydrated cement [32]. Moreover, the degree of C–S–H modification affected the morphology of the samples. With 5% C–S–H modification, the particles maintained a loosely stacked status. The particles adhered to each other and formed sheets as C–S–H content further increased, which is observed in the images of CC–CSH-10 and CC–CSH-20. The adhesion of CC–CSH with more C–S–H modification was in agreement with the increase in apparent size, as listed in Table 4.

EDS characterization was carried out with SEM imaging to verify whether CaCO_3_-C–S–H composite had formed and whether C–S–H deposited onto the CaCO_3_ surface. As its chemical nature indicated, the spectrum of CaCO_3_ showed a blend of the three elements: Ca, C, and O; no other elements were spotted. In CC–CSH, the growth of C–S–H nanogel on the CaCO_3_ surface was indicated by the presence of a Si signal. The intensity of the Si signal increased from 8% in CC–CSH-5 to 19% in CC–CSH-10, but remained stable in CC–CSH-20, indicating that the surface of CaCO_3_ has already been fully covered by C–S–H in CC–CSH-10. EDS spectra confirmed C–S–H deposition onto the CaCO_3_ surface.

### 3.2. Effect of CC–CSH on Mortar Strength

The compressive strength data of CC–CSH-modified mortars are shown in Figure 3. As can be observed, 12 h strength of the mortars showed a prominent increase that was clearly sequenced by the amount of C–S–H modified, in which CC–CSH-20 showed the highest increase of 128%. However, once free C–S–H was removed, the strengthening effect of the remaining CC–CSH notably weakened, but the corresponding mortar strength was still higher than that of CC–CSH-5, which may be attributed to the higher surface coverage. The strength increase in plain CaCO_3_ was insignificant at this age, which can be attributed to its unseeded surface. For mortars at 1 d, the strengthening effect of CC–CSH with higher C–S–H content was surpassed by CC–CSH-5; mortar with CC–CSH-5 admix had the highest strength increase of 27%. The better dispersity of CC–CSH-5 (as previous results suggest) may contribute to its better performance at this time point. Moreover, the effect of unmodified CaCO_3_ began to intensify. At the age of 7 d, mortars with CaCO_3_ and CC–CSH-5 admix had the best strength performance, while the strength enhancement of CC–CSH-10 and CC–CSH-20 continued to weaken. At 28 d, mortars with CaCO_3_ and CC–CSH-5 addition were still slightly higher in apparent value, while mortars with the other two CC–CSH admixes showed indifferent strength variation. However, the strength deviation of the 28 d samples is small and within the range of error overall. The better effect of CC–CSH and unmodified CaCO_3_ at 1–7 d may be due to the milder hydration acceleration effect and more adequate pace of hydration product build up, since excessively rapid growth of hydration products may lead to coarsening of the network [33]. Overall, the trend of compressive strength development showed notable differences among samples with different degrees of C–S–H modification, indicating the feasibility of tuning hydration acceleration efficiency of CaCO_3_ by C–S–H modification.

As for flexural strength, mortars with CaCO_3_ and CC–CSH admix at 12 h and 1 d experienced similar enhancing effects as compressive strength. However, 7 d mortars with CC–CSH-20 addition showed slightly inferior flexural strength than controls, indicating a more brittle nature of these mortars. This brittleness may also be caused by excessively fast hydration production formation in these samples [33], which is caused by the high C–S–H content of the two types of CC–CSH.

Considering the strength-increasing effect of CC–CSH at different ages, CC–CSH-5 showed the most balanced performance due to its considerable strength-enhancing effect at a very early age (12 h), though slightly weaker than CC–CSH of higher C–S–H content, and its good strength-enhancing effect at 1 d and 7 d, which is similar to CaCO_3_. It can be concluded that at this degree of C–S–H modification, CaCO_3_ and C–S–H exhibit synergistic effects on mortar strength.

The dosage dependence of CC–CSH-5 was then tested to determine the optimal range, with the results demonstrated in Figure 4. As can be seen, at 12 h, the strength also showed a positive correlation with dosage, but in later ages, the strength enhancement in mortars with the highest dosage (2.0%) soon weakened, just like the CC–CSH with high C–S–H contents. Mortar with 2.0% CC–CSH dosage also exhibited brittleness, with flexural strength slightly dropping. The overall performances of 0.6% and 1.2% CC–CSH dosage were the most satisfactory, indicating an optimal range roughly between 0.6% and 1.2%. The strength of CC–CSH-5-modified mortars showed the typical dosage dependence of nanomaterials [7,34], where the effectiveness was impeded at high dosages.

### 3.3. Effect of CC–CSH on Cement Hydration

To determine the effect of different C–S–H contents in CC–CSH on mortar strength from a more comprehensive prospective, SEM images of paste samples were taken in early (12 h) and late (28 d) ages, which are shown in Figure 5. As the images suggest, there were notably more hydration products on pastes with CC–CSH, and hydration products on CC–CSH-20 samples appeared coarser, which was probably caused by particle aggregation and fast hydration, as mentioned above. For samples at 28 d, the split surfaces of control paste and paste with CC–CSH-5 were denser, while a more cracked structure with coarse particles appeared in paste with CC–CSH-20 admix. The difference in quantity and morphology of hydration products at 12 h, as well as late age structure, confirmed our suggestions in previous sections: the excessive acceleration effect of CC–CSH with high C–S–H (10%, 20%) content promoted the formation of coarse hydration products and coarse structures, which may exert adverse effects on late-age strength.

For a more quantitative investigation of the hydration acceleration effect of CC–CSH, the hydration heat of the samples of the initial 36 h was measured, with the results shown in Figure 6. As Figure 6a illustrates, there was a clear leftward shift in the main hydration peak to earlier ages in all samples with CC–CSH, but the main hydration peak of the sample with CaCO_3_ showed a slight rightward trend in the first few hours, which may be due to the retarding effect from the polycarboxylate dispersant added during its preparation as a reference.

For different CC–CSH samples, the main peaks in curves of CC–CSH showed clear dependence on the degree of C–S–H modification, and the main peak moved more leftward as the degree increased. This confirms the tuning effect of C–S–H modification on CaCO_3_. Moreover, the acceleration effect of CC–CSH-10 and CC–CSH-20 may be excessively high so as to stiffen microstructure and induce the formation of coarse hydration products, which has adverse effect on later-age strengths. The earlier main peak also indicates an earlier deceleration stage, which may also contribute to the inferior 1 d and 7 d strength enhancement of CC–CSH-10 and CC–CSH-20, as compared to CC–CSH-5.

To quantitatively evaluate the acceleration effect of CC–CSH, an acceleration coefficient *A* was calculated using the following equation based on the method of Luc Nicoleau and co-workers [35,36]:*A* = *Acc*/*Acc*_ctrl_,(1)
where *Acc* is the slope of the ascending section of the main peak of a CC–CSH-modified sample, and *Acc*_ctrl_ is the slope of the ascending section of the main peak of the control sample. C (%) is the dosage of CC–CSH.

According to the results, *A* of CC–CSH (1.57–1.77) at 1.2% was considerably higher than CaCO_3_ at the same dose (1.19), which is due to the pre-seeding of C–S–H nanogels. In addition, A showed dependence on the degree of C–S–H modification, indicating the effect of C–S–H on tuning the properties and the hydration acceleration effect of CaCO_3_. The higher apparent *A* values of CC–CSH-10 and CC–CSH-20 were largely due to the free C–S–H in the dispersion. Once the free C–S–H was removed, a notable drop was observed. CC–CSH-20 without free C–S–H had a slightly lower *A*, which is clearly due to adhesion of the particles caused by excessive C–S–H. All IC test results were in agreement with previous results and offered further confirmation of our findings.

## 4. Conclusions

From the results above, several conclusions can be drawn:(1)C–S–H nanogels can be artificially grown on CaCO_3_ particles, achieved through the seeding effect of CaCO_3_. The amount of nanogels on the CaCO_3_ surface is restricted by the available surface. Excessive nanogels would exist in free form in solution. The maximum content of C–S–H on the surface is reached at the modification degree of 5–10% m (CaCO_3_).(2)The C–S–H nanogels on CaCO_3_ surface alters its interfacial properties and exhibit a tuning effect on the cement hydration. The effect strengthens with higher surface C–S–H coverage, and it was halted once surface coverage reached the maximum.(3)Pre-seeding CaCO_3_ with C–S–H can considerably enhance its strengthening effect on cement in early ages (<1 d). The enhancement is due to the better seeding effect on C–S–H. However, excessive C–S–H modification would weaken CC–CSH’s effect on mechanical properties at later ages (>1 d), which is caused by excessively fast hydration product growth and particle agglomeration. Considering strength at all ages, CC–CSH-5 showed the most balanced performance among all the CC–CSH samples. It can be inferred that CaCO_3_ and C–S–H exhibited synergistic effects at the modification degree of CC–CSH-5. According to the strength tests on dosage dependence, the optimal dosage of CC–CSH-5 is 0.6–1.2%.

From our findings, we can conclude that tuning the performance of nanomaterials in cementitious systems by building nanocomposites is viable. The effect of a nanomaterial can be tuned at a fixed dosage, which offers a novel route for the modification and performance tuning of cement-oriented nanomaterials. Still, some aspects of nanocomposite preparation and modification are worth noting: the degree of modification should be carefully controlled to achieve the maximal synergistic effect of the components, and this degree seems to be closely related to the interfacial properties of the components; in this study, it was the available surface of CaCO_3_ that set the optimal degree. In addition, further work is essential for the practical application of this technique, as well as to confirm its competitiveness against other techniques.

## Data Availability

Not applicable.

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
