# Peer review of "Tuning the Hydration Acceleration Efficiency of Calcium Carbonate by Pre-Seeding with Calcium Silicate Hydrate"

_materials, 2022, doi:10.3390/ma15196726_

Round 1

Reviewer 1 Report

Please carefully read the comments listed below and “fully” address them:

1-    The ABSTRACT is not written in a logical order. Start with an overview of the topic and a rationale for your paper. Describe the methodology you used and the general outline of the manuscript. Finally, state the results in more detail (i.e., provide some numbers).

2-    The seeding effect/mechanism of nanogels is not novel and has been discussed in the literature over the past 5-10 years. Therefore, the novelty of your work should be further explained to the reader. 

3-    “Further application of nanotechnology for cementitious 31 hydration acceleration may help relieve the limitations of current techniques, such as 32 strength loss and durability deterioration at later ages. [20]”. Please specifically name concrete durability issues, such as “Delayed Ettringite Formation”, “Unsoundness”, and “Alkali-Silica-Reaction”, and for each condition "reference" the papers listed below:

“Delayed Ettringite Formation”:

·      Paul, A., Rashidi, M., Kim, J. Y., Jacobs, L. J., & Kurtis, K. E. (2022). The impact of sulfate-and sulfide-bearing sand on delayed ettringite formation. Cement and Concrete Composites, 125, 104323.

“Unsoundness”:

·      Kabir, H., Hooton, R. D., & Popoff, N. J. (2020). Evaluation of cement soundness using the ASTM C151 autoclave expansion test. Cement and Concrete Research, 136, 106159. 

Alkali-Silica-Reaction”:

·      Li, Z., Thomas, R. J., & Peethamparan, S. (2019). Alkali-silica reactivity of alkali-activated concrete subjected to ASTM C 1293 and 1567 alkali-silica reactivity tests. Cement and Concrete Research, 123, 105796.

4-    Please further detail on how the PCE solution was created, and why the optimum 40% solution was made?

5-    In Table 1: please report the cement loss on ignition (LOI), Blaine Fineness, and other important information such as physical properties (e.g., strength development with time).

6-    Label all the components/elements of particles shown in Fig. 1 including free C-S-H. Increase the scale size in Fig. 2. Add an error bar to all the data shown in Figs. 3 and 4, and explain how many samples were tested in order to report the strength values shown in these figures. 

7-   “all samples with CC-CSH, but samples with CaCO3 showed retarding trend” please fully detail how the retarding effect was quantified, was it based on localizing the (second) peak of IsoCal curves?

8-    More explanation is needed to justify the optimum 5% CSH on the overall performance. Also, can the authors highlight future research directions and recommendations in the conclusions? In addition, highlight the assumptions and limitations (e.g., 1-2 shortcoming(s) of the present study)? Finally, recheck your manuscript and polish it for grammatical mistakes. 

Author Response

Point 1: The ABSTRACT is not written in a logical order. Start with an overview of the topic and a rationale for your paper. Describe the methodology you used and the general outline of the manuscript. Finally, state the results in more detail (i.e., provide some numbers).

Response 1: The Abstract has been rewritten and reorganized, thanks for the reviewer's suggestions.

Point 2: The seeding effect/mechanism of nanogels is not novel and has been discussed in the literature over the past 5-10 years. Therefore, the novelty of your work should be further explained to the reader. 

Response 2: The introduction has been revised (mainly in Par.3) with more details concerning recent trend and the purpose of building the composite material.

Point 3: “Further application of nanotechnology for cementitious 31 hydration acceleration may help relieve the limitations of current techniques, such as 32 strength loss and durability deterioration at later ages. [20]”. Please specifically name concrete durability issues, such as “Delayed Ettringite Formation”, “Unsoundness”, and “Alkali-Silica-Reaction”, and for each condition "reference" the papers listed below:

“Delayed Ettringite Formation”:

  • Paul, A., Rashidi, M., Kim, J. Y., Jacobs, L. J., & Kurtis, K. E. (2022). The impact of sulfate-and sulfide-bearing sand on delayed ettringite formation. Cement and Concrete Composites, 125, 104323.

“Unsoundness”:

  • Kabir, H., Hooton, R. D., & Popoff, N. J. (2020). Evaluation of cement soundness using the ASTM C151 autoclave expansion test. Cement and Concrete Research, 136, 106159. 

“Alkali-Silica-Reaction”:

  • Li, Z., Thomas, R. J., & Peethamparan, S. (2019). Alkali-silica reactivity of alkali-activated concrete subjected to ASTM C 1293 and 1567 alkali-silica reactivity tests. Cement and Concrete Research, 123, 105796.

Response 3: Relevant literature has also been referred.

Point 4: Please further detail on how the PCE solution was created, and why the optimum 40% solution was made?

Response 4: We are sorry that the PCE used in this study is a commercial product, thus detailed preparation process about this specific one is not openly available. Still, typical preparation procedures of a polycarboxylate type dispersant can be found in the literatures below:

M. Kinoshita, Y. Yuki, K. Saitou, T. Takahashi. Methacrylic type water soluble polymer as high-range water reducing agent for ultra high-strength concrete. Kobunshi Ronbunshu, 52 (6) (1995), pp. 357-363
H. Cho, T. Park, J. Suh. Physical properties and characteristics of a cement admixture of poly(carboxylate-g-ethyleneoxide) by changing the reaction temperature, the reaction time and the initiator concentration. The 15th Symposium on Chemical Engineering Kyushu (Japan)-Taejon/Chungnam, KICHE (Korea Institute of Chemical Engineering), Korea (2002), pp. 90-91

The concentration of 40% was chosen probably due to a balance on manufacturing efficiency (to produce more in one batch) and the ease to handle the polymerization process for desired structure and yield.

Point 5: In Table 1: please report the cement loss on ignition (LOI), Blaine Fineness, and other important information such as physical properties (e.g., strength development with time).

Response 5: More physiochemical details about the cement have been supplemented in Table 1.

Point 6: Label all the components/elements of particles shown in Fig. 1 including free C-S-H. Increase the scale size in Fig. 2. Add an error bar to all the data shown in Figs. 3 and 4, and explain how many samples were tested in order to report the strength values shown in these figures. 

Response 6: The figures have been revised according to the suggestions. repetition of samples in strength tests has been specified in Section 2.3 (reorganized with 2.4)

Point 7: “all samples with CC-CSH, but samples with CaCO3 showed retarding trend” please fully detail how the retarding effect was quantified, was it based on localizing the (second) peak of IsoCal curves?

Response 7: Yes, determination of whether an admixture is accelerating or retarding is based on the second(main) peak of IsoCal curve. Relevant discussion has been revised for clearer expression.

Point 8: More explanation is needed to justify the optimum 5% CSH on the overall performance. Also, can the authors highlight future research directions and recommendations in the conclusions? In addition, highlight the assumptions and limitations (e.g., 1-2 shortcoming(s) of the present study)? Finally, recheck your manuscript and polish it for grammatical mistakes. 

Response 8: The statement has been revised with more focus on balanced performance rather than "optimal", The conclusion section has been revised for more highlights on outlook and limitations. There have also been thorough checks on grammatical mistakes and relevant revisions. 

Reviewer 2 Report

The purpose of this paper is to study the effect of calcium carbonate CSH seeding in the hydration of cement. The topic is interesting and timely, since this technique has not yet been completely explored. The experimental program is satisfactory, with an appropriate number of case studies and assessed properties, but can be improved. The overall structure of the paper is appropriate, but some suggestions are given. Figures and tables also require a few alterations. Regarding the introduction section, it shows a poor structure, the ideas are little detailed, unclear and sometimes repeated. Also there is a big issue concerning the English writing, either in terms of grammar construction or in the way ideas are presented, sometimes compromising the understanding of the content. Sometimes the speech is dense, with an unclear structure. A language revision, preferably by a professional editing service, is advised. More detailed comments are given below, but not exhaustively. As consequence, in my opinion this paper should only be accepted after major revision.

-          Some suggestions of English writing improvement are given in the attached pdf.

-          Abstract should be restructured. Authors give many details on the methodology and conclusions at a moment when readers are not properly aware of the content of the work. Hence, readers cannot completely understand the text. Authors should summarize the methodology and conclusions to essential information and add contents on the motivation and objective of the work at the beginning.

-          Introduction: the acceleration effect on cement is not properly introduced, both I terms of relevance and working mechanisms (it is not mentioned that the hydration is a reaction of the cement compounds). Eg.: “In cement and concrete, nanomaterials are capable of hydration acceleration” it might suggest that are the nanomaterials that show the acceleration reaction. Other examples in the attached pdf.

-          Introduction also shoes many writing issues: spaces missing between word and reference; wrong punctuation.

-          Rows 27-31: too many ideas in the same sentence, unclear content, spaces missing between word and reference; wrong punctuation.

-          Rows 60, 65,68, 80, 89, 90, 96, 108, 129: verbs in the wrong tense.

-          Row 68: add standard clarifying/supporting the cement denomination.

-          Row 68, 274, 282: punctuation missing.

-          Rows 72, 90, 110, 163, 185: too many ideas in the same sentence and poor sentence construction.

-          Rows 75, 77, 82, 84, 104, 113: unclear text, please improve writing.

-          Row 86: replace 2.2.1 by 2.2.2.

-          Structure of section 2: 2.3 and 2.4 should have the same criterion to assign captions: 2.3 methods used for mortars - focus on the material under testing; 2.4 characterization of hydration process - focus on the purpose of the tests. As consequence, preparation of mortars and pastes that should be in the same section are in different sections.

-          Structure of section 2: Include table with the list of pastes and mortars tested in correspondence with Figures 3, 4 and 6; pastes and mortars should have a notation indicating if they are one or other.

-          Figure 4 regards mortars cc-csh-5 with different dosages: clarify the preparation and composition of these mortars; what is the dosage of cc-csh-5 in Figure 3?

-          Mortar preparation: how many repetitions were made in each batch?

-          Preparation of paste samples: please clarify the composition of the solid material; do not use the word “binder” to include all the solid materials; improve the structure of the paragraph stating all the information regarding the composition together at the beginning (part is in the beginning, part in the end, other information type in the middle).

-          Structure of section 2.4: as before, criterion for sections is not the same; 2.4.3 also regards a paste processing for characterization, the caption of 2.4.2.

-          Section 3.1 Preparation and characteristics of CC-CSH particles: preparation is not a result.

-          Row 146: replace Table 4 by Table 3.

-          Table 3: clarify if the % content is in terms of the entire amount of free nano CSH or in terms of the solid content.

-          Figure 3: identification of (w/o) is not included.

-          Row 199: Authors mention “at 28d mortars with CaCO3 and CC-CSH-5 addition were still slightly higher than control”. I have the following doubts here: how much higher? Figure 3 does not enable a proper discussion. The increment due to addition can be within the standard deviation; standard deviation should be included, as well as other way to better illustrate the variation in relation to standard (usually a chart with the relative variation in %).

-          Row 201 “…their milder hydration acceleration effect, which avoids formation of coarse and stiff hydration products”: very poor explanation; authors should compare with results obtained by other authors.

-          Row 202 “Overall, compressive strength results indicated that the hydration acceleration effect of CaCO3 can be tuned by C-S-H pre-seeding”: partially inaccurate; the effect is only visible before the 28 days.

-          Rows 212, 232: sentence requires support on references.

-          Row 225: include reference to Figure 4.

-          Figures: use the same criterion in the diverse figures, entire word or abbreviation.

-          Conclusions, row 299: include the optimal dosage.

-          More detailed remarks are included in the attached pdf.

Author Response

Point 1: Some suggestions of English writing improvement are given in the attached pdf.

-          Abstract should be restructured. Authors give many details on the methodology and conclusions at a moment when readers are not properly aware of the content of the work. Hence, readers cannot completely understand the text. Authors should summarize the methodology and conclusions to essential information and add contents on the motivation and objective of the work at the beginning.

Response 1: The abstract has been rewritten for clear demonstration of aim, methodology and key findings.

Point 2: Introduction: the acceleration effect on cement is not properly introduced, both I terms of relevance and working mechanisms (it is not mentioned that the hydration is a reaction of the cement compounds). Eg.: “In cement and concrete, nanomaterials are capable of hydration acceleration” it might suggest that are the nanomaterials that show the acceleration reaction. Other examples in the attached pdf.

-          Introduction also shoes many writing issues: spaces missing between word and reference; wrong punctuation.

Response 2: The Introduction has been revised for better logical demonstrstion.

Point 3: Rows 27-31: too many ideas in the same sentence, unclear content, spaces missing between word and reference; wrong punctuation.

-          Rows 60, 65,68, 80, 89, 90, 96, 108, 129: verbs in the wrong tense.

-          Row 68: add standard clarifying/supporting the cement denomination.

-          Row 68, 274, 282: punctuation missing.

-          Rows 72, 90, 110, 163, 185: too many ideas in the same sentence and poor sentence construction.

-          Rows 75, 77, 82, 84, 104, 113: unclear text, please improve writing.

-          Row 86: replace 2.2.1 by 2.2.2.

Response 3: The mention points has been revised.

Point 4: Structure of section 2: 2.3 and 2.4 should have the same criterion to assign captions: 2.3 methods used for mortars - focus on the material under testing; 2.4 characterization of hydration process - focus on the purpose of the tests. As consequence, preparation of mortars and pastes that should be in the same section are in different sections.

-          Structure of section 2: Include table with the list of pastes and mortars tested in correspondence with Figures 3, 4 and 6; pastes and mortars should have a notation indicating if they are one or other.

Response 4: The methodology section has been re-organized.

Point 5: Figure 4 regards mortars cc-csh-5 with different dosages: clarify the preparation and composition of these mortars; what is the dosage of cc-csh-5 in Figure 3?

Response 5: Relevant expression in methodology section has been revised, captions were also revised for better interpretation.

Point 6: Mortar preparation: how many repetitions were made in each batch?

Response 6: Details about repetition has been added.

Point 7: Preparation of paste samples: please clarify the composition of the solid material; do not use the word “binder” to include all the solid materials; improve the structure of the paragraph stating all the information regarding the composition together at the beginning (part is in the beginning, part in the end, other information type in the middle).

Response 7: Relevant section has been re-written.

Point 8: Structure of section 2.4: as before, criterion for sections is not the same; 2.4.3 also regards a paste processing for characterization, the caption of 2.4.2.

Response 8: 2.3 and 2.4 has been re-organized.

Point 9: Section 3.1 Preparation and characteristics of CC-CSH particles: preparation is not a result.

Response 9: The title has been revised

Point 10: Row 146: replace Table 4 by Table 3.

Response 10: Table and figure referring in the manuscript has been unified

Point 11: Table 3: clarify if the % content is in terms of the entire amount of free nano CSH or in terms of the solid content.

Response 11: The issue is clarified in the table note

Point 12: Figure 3: identification of (w/o) is not included.

Response 12: The un-pre-stated abbreviation (w/o) was replaced with full text in relevant figures.

Point 13: Row 199: Authors mention “at 28d mortars with CaCO3 and CC-CSH-5 addition were still slightly higher than control”. I have the following doubts here: how much higher? Figure 3 does not enable a proper discussion. The increment due to addition can be within the standard deviation; standard deviation should be included, as well as other way to better illustrate the variation in relation to standard (usually a chart with the relative variation in %).

Response 13: Relevant discussion has been revised.

Point 14: Row 201 “…their milder hydration acceleration effect, which avoids formation of coarse and stiff hydration products”: very poor explanation; authors should compare with results obtained by other authors.

Response 14: Discussion has been revised, literatures has been supplemented.

Point 15: Row 202 “Overall, compressive strength results indicated that the hydration acceleration effect of CaCO3 can be tuned by C-S-H pre-seeding”: partially inaccurate; the effect is only visible before the 28 days.

Response 15: The discussion has been revised to focus on comparing the difference between the samples.

Point 16: Rows 212, 232: sentence requires support on references.

Response 16: references has been added.

Point 17: Row 225: include reference to Figure 4.

Response 17: Figure reference has been included.

Point 18: Figures: use the same criterion in the diverse figures, entire word or abbreviation.

Response 18: The criteria has been unified.

Point 19: Conclusions, row 299: include the optimal dosage.

Response 19: Optimal dosage has been included.

Point 20: More detailed remarks are included in the attached pdf.

Response 20: Revisions have been made with suggestions in the attachment.

Round 2

Reviewer 1 Report

The authors addressed my comments, and the manuscript can be published in the present format. 

Reviewer 2 Report

The authors have improved the manuscript  and satisfactorily addressed most of my comments.